# Maternal Body Mass Index Is Associated with Profile Variation in Circulating MicroRNAs at First Trimester of Pregnancy

**DOI:** 10.3390/biomedicines10071726

**Published:** 2022-07-18

**Authors:** Kathrine Thibeault, Cécilia Légaré, Véronique Desgagné, Frédérique White, Andrée-Anne Clément, Michelle S. Scott, Pierre-Étienne Jacques, Renée Guérin, Patrice Perron, Marie-France Hivert, Luigi Bouchard

**Affiliations:** 1Department of Biochemistry and Functional Genomics, Faculty of Medicine and Health Sciences (FMHS), Université de Sherbrooke, Sherbrooke, QC J1K 2R1, Canada; kathrine.thibeault@usherbrooke.ca (K.T.); cecilia.legare@usherbrooke.ca (C.L.); veronique.desgagne@usherbrooke.ca (V.D.); andree-anne.clement@usherbrooke.ca (A.-A.C.); michelle.scott@usherbrooke.ca (M.S.S.); renee.guerin2@usherbrooke.ca (R.G.); 2Clinical Department of Laboratory Medicine, Centre Intégré Universitaire de Santé et de Services Sociaux (CIUSSS) du Saguenay-Lac-Saint-Jean—Hôpital de Chicoutimi, Saguenay, QC G7H 5H6, Canada; 3Département de Biologie, Faculté des Science, Université de Sherbrooke, Sherbrooke, QC J1K 2R1, Canada; frederique.white@usherbrooke.ca (F.W.); pierre-etienne.jacques@usherbrooke.ca (P.-É.J.); 4Department of Medicine, FMHS, Université de Sherbrooke, Sherbrooke, QC J1K 2R1, Canada; patrice.perron@usherbrooke.ca; 5Centre de Recherche du Centre Hospitalier Universitaire de Sherbrooke (CR-CHUS), Université de Sherbrooke, Sherbrooke, QC J1H 5N4, Canada; mhivert@partners.org; 6Department of Population Medicine, Harvard Medical School, Harvard Pilgrim Health Care Institute, Boston, MA 02115, USA; 7Diabetes Unit, Massachusetts General Hospital, Boston, MA 02114, USA

**Keywords:** microRNA, obesity, pregnancy, next-generation sequencing

## Abstract

Many women enter pregnancy with overweight and obesity, which are associated with complications for both the expectant mother and her child. MicroRNAs (miRNAs) are short non-coding RNAs that regulate many biological processes, including energy metabolism. Our study aimed to identify first trimester plasmatic miRNAs associated with maternal body mass index (BMI) in early pregnancy. We sequenced a total of 658 plasma samples collected between the 4th and 16th week of pregnancy from two independent prospective birth cohorts (Gen3G and 3D). In each cohort, we assessed associations between early pregnancy maternal BMI and plasmatic miRNAs using DESeq2 R package, adjusting for sequencing run and lane, gestational age, maternal age at the first trimester of pregnancy and parity. A total of 38 miRNAs were associated (FDR q < 0.05) with BMI in the Gen3G cohort and were replicated (direction and magnitude of the fold change) in the 3D cohort, including 22 with a nominal *p*-value < 0.05. Some of these miRNAs were enriched in fatty acid metabolism-related pathways. We identified first trimester plasmatic miRNAs associated with maternal BMI. These miRNAs potentially regulate fatty acid metabolism-related pathways, supporting the hypothesis of their potential contribution to energy metabolism regulation in early pregnancy.

## 1. Introduction

Overweight and obesity (OW/O) are worldwide health issues [1] increasingly affecting women of reproductive age [2,3] and are considered the most common health conditions in pregnancy [4]. The prevalence of obesity among women of reproductive age is 17.8% in Canada [5] and 39.7% in the United States [6]. Maternal OW/O can have serious short- and long-term consequences on the health of the mother and her child. OW/O in pregnancy are associated with an increased risk of gestational diabetes mellitus (GDM), gestational hypertension, preeclampsia (PE) and cesarean delivery [4]. Offspring exposed to maternal OW/O are more likely to suffer from macrosomia and neonatal hyperinsulinemic hypoglycemia [7]. Possibly through fetal metabolic programming (based on the Developmental Origin of Health and Disease-DOHaD), these offspring are also at increased risk of childhood obesity, type 2 diabetes, cardiovascular disease, asthma and neurodevelopmental disorders [8,9]. A better understanding of the pathophysiological mechanisms leading to pregnancy complications associated with maternal OW/O could lead to strategies to prevent to maternal morbidity and possibly lower their consequences on the exposed offspring.

MicroRNAs (miRNAs) are short non-coding, single-stranded, RNA molecules of 19 to 25 nucleotides. They regulate many biological processes by targeting messenger RNAs (mRNAs), leading to a decrease in protein synthesis [10]. Three miRNA clusters are predominantly expressed in the placenta: chromosome 19 miRNA (C19MC), chromosome 14 miRNA (C14MC) and miR-371-3 miRNA clusters. The abundance of miRNAs from C19MC gradually increases in maternity from conception until the end of pregnancy, whereas miRNAs from the C14MC cluster are more abundant at the beginning of pregnancy and their blood levels decrease throughout pregnancy [11,12]. These placental miRNAs may participate in the regulation of maternal physiology by intercellular communication through placental extracellular vesicle secretion in maternal circulation [13]. Indeed, miRNAs are suspected of having an important role in pregnancy, its maintenance as well as adaptation to its very specific physiologic needs [10,11]. Accordingly, dysregulation of the C19MC and C14MC has been associated with pregnancy complications, including PE, intrauterine growth restriction and insulin sensitivity regulation [13,14,15,16,17].

Until now, only a few studies have investigated miRNAs in pregnancy complicated by OW/O [18,19,20,21]. However, none of these studies were done by next-generation sequencing or in plasma in the first trimester of pregnancy. We hypothesized that maternal BMI at the first trimester of pregnancy is associated with variations in circulating levels of miRNAs. Therefore, our objectives were to identify plasmatic miRNAs associated with maternal BMI and the metabolic pathways they potentially regulate.

## 2. Materials and Methods

### 2.1. Discovery Cohort: Genetics of Glucose Regulation in Gestation and Growth (Gen3G) Cohort

We selected participants for the discovery step of the study from the Gen3G prospective pregnancy and birth cohort [22]. Briefly, we recruited women in the first trimester of pregnancy (between the 4th and the 16th week), and we followed them until delivery. An oral glucose tolerance test (OGTT—75 g) was performed between the 24th and 28th week of pregnancy for 854 women. For this study, our selection criteria were: women of European descent, 18 years old and older, not taking any medication that influences glycemia, free from pre-gestational diabetes, having singleton pregnancy as well as the availability of a plasma sample at the first trimester of pregnancy (500 µL), anthropometric measures (e.g., maternal BMI), follow-ups of the offspring at 3 and 5 years old, and genetics (Mother: Infinium MEGAEX Array, Illumina; Offspring: whole genome sequencing) and epigenetics (EPIC array) data. A total of 444 women fulfilled these criteria. Before the study began, consent was obtained from all participating women, and all protocols were approved by the Centre Hospitalier Universitaire de Sherbrooke (CHUS) ethics committee.

### 2.2. Anthropometric Measurements in Gen3G

BMI is used to classify individuals by dividing their weight (kg) by their height (m) squared. This measurement was taken in the first trimester of pregnancy (between the 4th and 16th week) [23], and a description of the BMI measurement has been published [22]. Briefly, weight was measured in kg on a calibrated electronic scale, and height was measured in meters with a wall stadiometer (without shoes). The BMI value was calculated from these data.

### 2.3. Replication Cohort: Design, Develop, Discover (3D) Cohort Study

A replication analysis was run on 226 participants selected from the 3D prospective birth cohort [24]. Briefly, the participants were recruited during their first trimester of pregnancy in nine different sites across the province of Québec, Canada. For this study, women of European descent with plasma samples (500 µL) at the 1st trimester of pregnancy and at least fasting and the 2 h post-OGTT glucose levels measured between the 24th and 28th week of pregnancy were included. Exclusion criteria included pre-existing diabetes, GDM diagnosed at the first trimester of pregnancy, chronic hypertension, or gestational hypertension diagnosed at the first trimester of pregnancy (Appendix A). Biological specimens (e.g., plasma) and anthropometric measurements were collected (measured by research staff). BMI was measured between the 8th and 14th week of pregnancy during the first trimester of pregnancy. Women without BMI data were also excluded. All women gave their informed consent before the beginning of the study according to the Helsinki declaration, and the protocols have been accepted by the ethical committees

### 2.4. RNA Extraction

We extracted total RNA with the MirVana PARIS kit (ThermoFisher Scientific, Waltham, United States, catalog #AM1556) from 500 µL of plasma collected between the 4th and 16th week of pregnancy following the standard protocol and eluted in 75 µL of nuclease-free water, in random order. We concentrated our RNA samples following the protocol established by Burgos et al. [25]. In brief, RNA was mixed with 35 µL of cold (4 °C) 7 M ammonium acetate solution (ThermoFisher Scientific, Waltham, United States, catalog #02002268) and mixed with 420 µL of chilled (−20 °C) absolute ethanol (Commercial Alcohols, ON, Canada; catalog #P006EAAN). RNA was precipitated overnight at −20 °C and centrifuged at 16,000× *g* at 4 °C for 30 min. The RNA pellet was washed twice with 200 µL of 80% ethanol and then centrifuged at 16,000× *g*, 4 °C for 5 min. The RNA pellet was dried for 30 min at room temperature and resuspended in 5 µL of nuclease-free water.

### 2.5. Library Preparation

We used the Truseq Small RNA Sample Prep kit (Illumina, BC, Canada, catalog #RS-200-0012) for library preparation. Concentrated RNA samples (5 µL) were randomly treated following the standard protocol adapted by Burgos et al. [25]. Briefly, half of the reagents were used for ligation of RNA at the 3′ and 5′ ends, reverse transcription, indexing (1–48, one index per sample) and PCR amplification (15 cycles), to maintain an optimal ratio between RNA and reagents. The libraries were purified by migration on a Novex polyacrylamide TBE Gel, 6% (ThermoFisher Scientific, Waltham, United States, catalog #EC265BOX) by selecting bands between 145–160 bp, eluted in 300 µL of nuclease-free water, and incubated overnight at room temperature and 500 RPM on an incubating microplate shaker (VWR, ON, Canada, catalog #12620-930). Then, the libraries were concentrated by precipitation following a standardized procedure (including a 30 min incubation of the precipitation mix at −80 °C), and the cDNA pellet was suspended in 25 μL of 10 mM Tris-HCI pH 8.5 buffer.

### 2.6. Library Quality Control and Sequencing

The libraries were sequenced at the McGill University and Génome Québec Innovation Centre (Montréal, Canada), either on a HiSeq 2500 or HiSeq 4000 platform (50 cycles, with 7 cycles indexing read) for the Gen3G samples. Twelve samples were extracted twice and sequenced on both the HiSeq 2500 and HiSeq 4000 platforms that we leveraged during our QC and normalization process to take into account potential technical and batch effects from the different sequencing platforms. Overall, Pearson correlation coefficients between miRNA levels measured on the 2 platforms for these 12 samples were ≥0.94. The data from both platforms were then combined for processing and statistical analysis but adjusted for run and lane as normally recommended [12]. Libraries were quantified by qPCR, equimolarly pooled (HiSeq 2500: 12 libraries with different indexes per lane at a molarity of 7 pM; HiSeq 4000: 20 libraries with different indexes per lane at a molarity of 10 pM), denatured and clustered on single-read Illumina flowcells (catalog # GD-401-3001 and catalog GD-410-1001) according to the manufacturer’s standard protocol.

The 3D replication study samples were sequenced on the NovaSeq 6000 platform at the McGill University and Génome Québec Innovation Centre (Montréal, Canada). Libraries were prepared following the procedure applied to Gen3G samples and quantified using the Kapa Illumina GA with Revised Primers-SYBR Fast Universal kit (Kapa Biosystems, Wilmington, United States), normalized, equimolarly pooled (48 libraries with different indexes per lane at a molarity of 225 pM), denatured and clustered on an Illumina NovaSeq S1 flowcell following the Xp protocol from the manufacturer’s recommendations. The run was performed for 100 cycles in single-end mode.

### 2.7. Bioinformatics Analysis

We first applied the extracellular RNA processing toolkit (exceRpt) pipeline (version 4.6) [26]. Briefly, exceRpt uses FASTX-Toolkit and FastQC to assess sequencing data quality after removing the adapters and poor quality (Phred score < 20 for 80% or more of the read) sequences. The remaining reads were then aligned to the human genome (GRCh37) and miRbase [27] (version 21) using STAR [28] (version 2.4.2a) alignment algorithm. After performing data visualization of the raw read counts, 9 outliers were excluded from the Gen3G cohort, and 3 outliers were excluded from the 3D cohort.

### 2.8. Statistical Analysis

Since all the participant characteristics (Gen3G and 3D cohorts) were not normally distributed based on a Shapiro-Wilk test, non-parametric Mann-Whitney U tests were applied to compare the two cohorts. Association between plasmatic miRNA levels and maternal BMI at the first trimester of pregnancy was assessed using the default parameters in DESeq2 R package [29]. The duplicate samples (*n* = 12) were combined using the collapseReplicates function from DESeq2 package. The statistical model was adjusted for sequencing run and lane, gestational age, maternal age at the first trimester of pregnancy, and parity. A sensitivity analysis was also performed by adding the fetal sex as a covariate to the analysis model. The results remain overall unchanged. MiRNAs were considered significantly associated with maternal BMI with a false discovery rate (FDR) adjusted q-value < 0.05, where the fold change represents the change in the miRNAs normalized read counts per unit of BMI. The EnhancedVolcano [30] package was used for the creation of the volcano plot. For replication, criteria were first the direction of the associations, the fold changes, and finally the nominal *p*-values (one-tailed). The statistical analyses were all done with R version 4.0.3 in R studio version 1.4.1103.

### 2.9. Biological Pathway Analysis

The potential biological function of the maternal BMI-associated miRNAs was evaluated using miRPath v.3 software from DIANA tools [31]. In brief, Kyoto Encyclopedia of Genes and Genomes (KEGG) metabolic pathway analysis was done on all the miRNAs that were significantly (q-values < 0.05) associated with maternal BMI at the first trimester of pregnancy and replicated for their fold change value in the 3D cohort. Tarbase 7.0 was selected for miRNA-mRNA interactions as it considers those validated experimentally. The pathway union parameter was used to merge the results with the Fisher exact test enrichment analysis method. The default settings in miRPath v.3 and FDR correction were applied, and the q-value threshold was set to 0.05.

## 3. Results

### 3.1. Participant’s Characteristics

Table 1 presents the characteristic of the selected Gen3G and 3D cohorts’ participants. Women from Gen3G were on average 28.48 ± 4.26 years old, had a mean BMI of 25.95 ± 5.98 kg/m^2^ at the first trimester of pregnancy (mean gestational age: 9.63 ± 2.26 weeks), and had a mean parity of 0.69 ± 0.91 (53.56% nulliparous). On average, women from the 3D cohort were slightly older (30.60 ± 3.90 years old, *p* < 0.001) and had a greater gestational age in the 1st trimester visit (11.89 ± 1.52 weeks, *p* < 0.001) and a lower parity (0.47 ± 0.66, *p* = 0.01; 60.99% nulliparous), whereas their mean BMI (25.39 ± 5.91 kg/m^2^, *p* = 0.22) was similar to that of the women from the Gen3G cohort.

### 3.2. Association between miRNA Levels and Maternal BMI at the First Trimester of Pregnancy

In Gen3G, 2170 miRNAs were expressed in the plasma of pregnant women. A miRNA was considered detected when it has >1 read per participant. A total of 61 miRNAs (FDR-adjusted q-values < 0.05) were associated with maternal BMI in the first trimester of pregnancy (Figure 1). The list of significant miRNAs with their mean normalized read counts, fold changes (FC), unadjusted *p*-values, and FDR-adjusted q-values are shown in Appendix A. Among these 61 miRNAs, higher BMI was associated with lower circulating levels for 48 miRNAs and with higher circulating levels for 13 miRNAs. Interestingly, 28 of these miRNAs (46%) are encoded by the C19MC, 1 miRNA by the C14MC, and 3 miRNAs by the miR-371-3 miRNAs cluster. Table 2 shows the top 10 miRNAs with the most significant associations with maternal BMI; for these 10 miRNAs, higher BMI was associated with lower circulating levels.

### 3.3. Replication of miRNAs Associated with Maternal BMI at the First Trimester of Pregnancy in the 3D Cohort

We conducted replication analyses for the 61 identified miRNAs (with q < 0.05) in 3D cohort (see Appendix A). Interestingly, we achieved full replication based on the strength of the associations (one tailed *p*-value < 0.05) as well as the direction and magnitude of the fold changes between Gen3G and 3D cohorts for 22 (36%) miRNAs (Table 3). Moreover, 8 (80%) out of the 10 most strongly associated miRNAs in Gen3G cohort were replicated in 3D (*p* < 0.05). Sixteen (16) additional miRNAs (total 38) had a similar fold change (direction and magnitude) between the Gen3G and 3D cohort without reaching our a priori selected statistical significance threshold (Appendix A). These miRNAs were included in the pathway analysis below.

### 3.4. Metabolic Pathway Analysis of miRNAs Associated with Maternal BMI at the First Trimester of Pregnancy

To assess the potential biological role of the 38 miRNAs associated with maternal BMI at the first trimester of pregnancy, we did a metabolic pathway analysis using miRPath v3 [32]. Seven KEGG pathways from the union pathway analysis were enriched with targets of these miRNAs. Figure 2 shows these targeted metabolic pathways and their FDR-adjusted q-value. Interestingly, the top 2 pathways were related to fatty acid metabolism (*p* = 1 × 10−325, 6 miRNAs) and fatty acids biosynthesis (*p* < 1 × 10−325, 4 miRNAs).

## 4. Discussion

Many women enter pregnancy with OW/O, which are associated with complications, such as GDM and pre-eclampsia. In the current study, we sequenced 658 plasma samples from two independent birth cohorts and identified over 20 miRNAs circulating in maternal plasma at first trimester that are associated with maternal BMI in early pregnancy. To the best of our knowledge, this is the largest study using next generation sequencing to identify miRNAs at the first trimester of pregnancy associated with maternal BMI.

Only a few studies have associated maternal OW/O in pregnancy with plasmatic miRNA profile dysregulation [15,16,18,19]. Interestingly, four of our top 10 miRNAs, all replicated in 3D, were previously associated with maternal obesity in cord blood [18]. Jing et al. reported that cord blood hsa-miR-1323, hsa-miR-516b-5p, hsa-miR-516a-5p, and hsa-miR-520a-3p levels were positively associated with maternal OW/O [18], which is contrary to what we have observed in the current study. This apparent discrepancy could be explained by the difference in sample origin (fetal whole cord blood vs. maternal plasma) and the timing during pregnancy (1st trimester vs. at delivery). These four miRNAs are all encoded in the C19MC, the expression of which is known to increase throughout pregnancy. Overall, our results suggest that the impact of OW/O might be dependent on the timing of collection and the tissue tested (maternal plasma vs. cord blood). In addition to these four miRNAs, our study also identified novel miRNAs associated with maternal BMI, of which many are also encoded by the C19MC. None of our miRNAs were reported in the other three studies [15,16,19].

One fascinating hypothesis is that the placenta secretes miRNAs into maternal blood contributing to feto-maternal communication and metabolic adaptation throughout pregnancy [32]. Indeed, more than 600 miRNAs dynamically expressed in the human placenta have been reported so far [10]. From these, 127 miRNAs are encoded into the C14MC, C19MC and miR-371-3 clusters. [10,11] Interestingly, 17 of the 22 (77%) replicated miRNAs are encoded by the C19MC. More broadly, 21 out of the 38 replicated miRNAs (55%) are from the C19MC, 1 from the C14MC, and 2 from the miR-371-3. C19MC is a large miRNA cluster specific to primates that encodes 46 miRNAs genes, 58 mature miRNAs and is maternally imprinted (paternally expressed) [10]. These miRNAs are also strongly expressed in trophoblasts, suggesting an important role in pregnancy and embryonic and fetal development [10]. In a previous study, we also identified miRNAs specific to pregnancy (pregnant vs. non-pregnant women) and varying between the 4th and the 16th week of pregnancy, using the same Gen3G microtranscriptomic dataset [12]. Accordingly, 5 more miRNAs (in addition to the 8 from the C19MC) of our replicated miRNAs were found upregulated in pregnancy in that previous study. This provides novel support of the roles of plasma miRNAs in pregnancy. Specifically, maternal BMI levels in pregnancy may influence the expression of miRNAs that are more abundant in pregnancy and may thus have a possible role in metabolic adaptation and development.

Maternal OW/O are also risk factors for GDM and PE and consequently the miRNAs we have identified could also contribute to the development of these complications. In another previous study from our group and using the same Gen3G microtranscriptomic dataset, we identified miRNAs in the first trimester of pregnancy associated with and predictive of insulin sensitivity in the second trimester of pregnancy. Among our replicated miRNAs, 15 were also associated with insulin sensitivity [17]. Similar analyses were also conducted to identify miRNAs associated and predictive of GDM still using this microtranscriptomics and Gen3G datasets. Among the replicated miRNAs, 16 were also associated with GDM [33]. Overall, 13 were associated with maternal obesity, insulin sensitivity and GDM. Moreover, higher circulating levels of serum hsa-miR-1323 has been associated with GDM [34], hsa-miR-517-5p and hsa-miR-520a-5p were reported to be downregulated in PE, which is consistent with our results with maternal BMI [16,35]. Finally, lower circulating levels of hsa-miR-526b-5p were detected in patients with metabolic syndrome – characterized by central excess adiposity—a direction of association that is concordant with our results [36]. Through their mechanism of action, the miRNAs that we found associated with maternal BMI might potentially have a role to play in the pathophysiology of how excess weight in early gestation may lead to pregnancy complications. However, further studies are needed to disentangle their specific roles and the causal relationship.

Our pathway analysis showed that the miRNAs associated with maternal BMI regulate a few metabolic pathways, of which lipid metabolism is of high interest for its biologic relevance. Overall, 10 miRNAs were linked to fatty acid biosynthesis (5 miRNAs), fatty acid metabolism (10 miRNAs), or fatty acid elongation (5 miRNAs) pathways. They were either up—(*n* = 5) or down—(*n* = 5) regulated in relation to maternal BMI. Obesity, inside and outside of pregnancy, is associated with dyslipidemia, among several other metabolic complications [37]. Early pregnancy is characterized by a lipogenic profile which is believed to favor energy storage that will be later used to meet the metabolic demand of both the mother and her growing fetus during pregnancy [38]. By targeting fatty acid biosynthesis and metabolism, our BMI-associated miRNAs could play an early role in the regulation of the lipid metabolism pathways in pregnancy or dysregulation in response to maternal OW/O. Further studies are also needed to understand the roles of plasma miRNAs in metabolic adaptation to pregnancy and their links with maternal OW/O.

### Strengths and Limitations

To the best of our knowledge, this is the largest study investigating circulating miRNAs associated with maternal BMI in the first trimester of pregnancy using next-generation sequencing. Among the strengths, our study was conducted in two large, well-characterized prospective pregnancy and birth cohorts and on plasma samples collected early in pregnancy, from the 4th to 16th weeks. This allowed us to assess obesity-related miRNA dysregulation months before pregnancy complications, such as GDM and PE, develop. Finally, we applied next generation sequencing technology, which allowed us to fully profile and quantify plasma miRNAs. This technology is a robust and sensitive approach for miRNA quantification [39]. Finally, although our study was by design associative, the independent replication significantly improves the robustness of the results we are reporting.

Our study also has some limitations. First, BMI is an easy and convenient assessment of OW/O but remains a surrogate measure of fat mass. Nevertheless, BMI remains recommended by the WHO to assess OW/O. Also, our BMIs were measured at the time plasma samples were collected. Overall, BMI in the first trimester of pregnancy is considered an acceptable assessment of that before pregnancy as weight is relatively stable in the first weeks. Also, it seems clear that functional studies are needed to confirm the role of these miRNAs associated during pregnancy (metabolic adaptation, fetal development and in pregnancy complications.)

## 5. Conclusions

We have identified 22 plasma miRNAs associated with maternal BMI in the first trimester of pregnancy. Identified miRNAs were enriched in biological pathways related to fatty acids and lipids that are implicated in the pathophysiology of OW/O and mostly encoded by the C19MC, which is mainly expressed by the placenta. These results could provide new insights into the understanding of the effect of OW/O on the development of different complications in pregnancy such as GDM and PE.

## Figures and Tables

**Figure 1 biomedicines-10-01726-f001:**
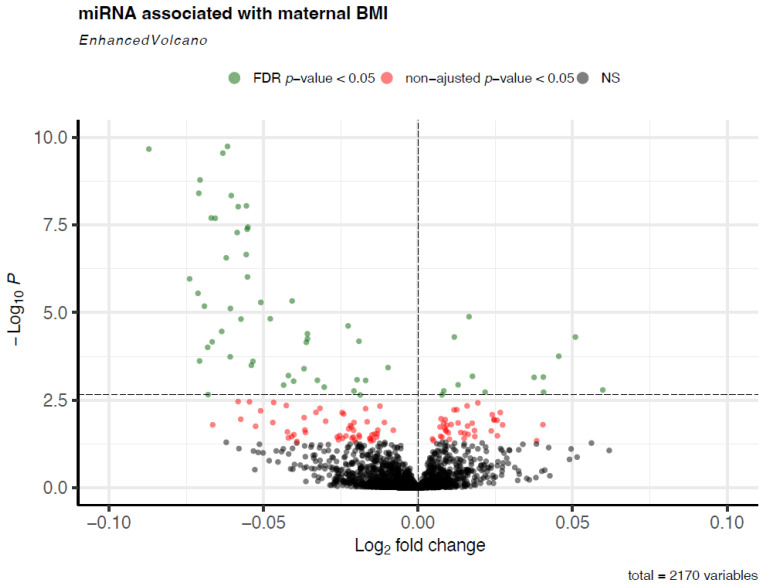
Plasmatic miRNAs associated with maternal BMI at 1st trimester of pregnancy. This volcano plot shows the miRNAs associated with maternal BMI. Each point represents a single miRNA. The vertical dotted line represents a log2 fold change of 0, and the horizontal dotted line represents the FDR-adjusted q-value threshold of < 0.05. The model was adjusted for sequencing lane and run, maternal and gestational age at 1st trimester, and parity. The fold change represents the change in miRNA abundance for an increase of one unit of maternal BMI in 1st trimester of pregnancy. Abbreviations: FDR: false discovery rate; NS: not significant.

**Figure 2 biomedicines-10-01726-f002:**
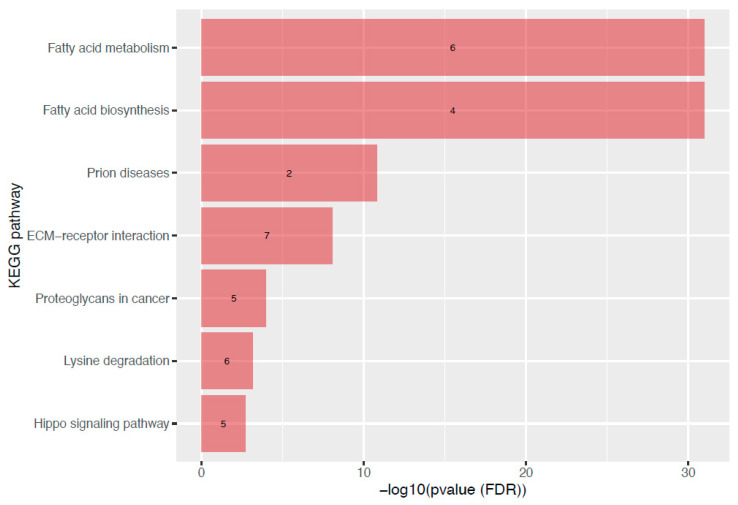
KEGG pathways targeted by miRNAs associated with maternal BMI at the first trimester of pregnancy. This bar graph is showing the KEGG pathways enriched with targets of miRNAs associated with maternal BMI. Each bar represents one pathway, ranked according to its level of significance (FDR adjusted q-value). The number of miRNAs involved in the pathway is shown directly in the bar. Abbreviations: ECM: extracellular matrix; FDR: false discovery rate; KEGG: Kyoto Encyclopedia of Genes and Genomes.

**Table 1 biomedicines-10-01726-t001:** Maternal characteristics at the first trimester of pregnancy in Gen3G (*n* = 435) and 3D (*n* = 223) cohorts.

Characteristics	Gen3G Cohort		3D Cohort		Comparison between Cohorts (*p*-Value) *
	Mean ± SD	Range	Mean ± SD	Range	
**BMI (kg/m^2^)**	25.95 ± 5.98	16.10–54.10	25.39 ± 5.91	16.80–48.50	0.22
**Age (years)**	28.48 ± 4.26	18–47	30.60 ± 3.90	20–42	<0.001
**Gestational age (weeks)**	9.63 ± 2.26	4.09–16.30	11.89 ± 1.52	7.71–16.43	<0.001
**Parity** **(% nulliparous)**	0.69 ± 0.91 (53.56)	0–6	0.47 ± 0.66 (60.99)	0–3	0.01

* Comparisons between Gen3G and 3D cohorts were performed using a Mann-Whitney U test. Abbreviations: 3D: Design, Develop, Discover birth cohort; BMI: body mass index; Gen3G: Genetics of Glucose regulation in Gestation and Growth birth cohort; SD: standard deviation.

**Table 2 biomedicines-10-01726-t002:** Top 10 miRNAs most significantly associated with maternal BMI at first trimester of pregnancy in the Gen3G cohort.

miRNA	DESeq2 Normalized Read Count (Mean ± SD)	Fold Change *	Unadjusted *p*-Value	FDR-Adjusted q-Value
hsa-miR-1323 ^a^	146.39 ± 230.60	0.957	2.79 × 10^−^^10^	9.60 × 10^−8^
hsa-miR-516b-5p ^a^	101.50 ± 150.35	0.958	1.79 × 10^−^^10^	9.60 × 10^−8^
hsa-miR-371a-5p ^b^	11.02 ± 17.09	0.941	2.13 × 10^−^^10^	9.60 × 10^−8^
hsa-miR-525-5p ^a^	9.07 ± 15.31	0.952	1.63 × 10^−9^	4.21 × 10^−7^
hsa-miR-516a-5p ^a^	31.25 ± 53.81	0.959	4.55 × 10^−9^	7.83 × 10^−7^
hsa-miR-524-5p ^a^	8.64 ± 14.48	0.952	3.90 × 10^−9^	7.83 × 10^−7^
hsa-miR-518e-5p ^a^	43.96 ± 62.52	0.962	8.92 × 10^−9^	1.22 × 10^−6^
hsa-miR-520a-3p ^a^	86.75 ± 132.37	0.960	9.41 × 10^−9^	1.22 × 10^−6^
hsa-miR-518e-3p ^a^	7.99 ± 14.22	0.956	2.02 × 10^−8^	2.09 × 10^−6^
hsa-miR-520d-5p ^a^	5.92 ± 10.35	0.955	1.98 × 10^−8^	2.09 × 10^−6^

Model adjusted for sequencing run and lane, gestational age, maternal age at the first trimester of pregnancy, and parity. * Fold changes represent the change in miRNA abundance for each increase of one unit of maternal BMI at 1st trimester of pregnancy. ^a^ miRNAs from C19MC. ^b^ miRNAs from miR-371-3 miRNAs cluster. Abbreviations: FDR: false discovery rate; SD: standard deviation.

**Table 3 biomedicines-10-01726-t003:** miRNAs significantly associated with maternal BMI at 1st trimester of pregnancy in Gen3G cohort and fully replicated in 3D cohort.

	Gen3G				3D		
miRNA	DESeq2 Normalized Read Count (Mean ± SD)	Fold Change *	Unadjusted *p*-Value	FDR-Adjusted q-Value	DESeq2 Normalized Read Count (Mean ± SD)	Fold Change *	Unadjusted *p*-Value
hsa-miR-1323 ^a^	146.39 ± 230.60	0.957	2.79 × 10^−^^10^	9.60 × 10^−8^	676.45 ± 581.88	0.966	2.56 × 10^−5^
hsa-miR-516b-5p ^a^	101.50 ± 150.35	0.958	1.79 × 10^−^^10^	9.60 × 10^−8^	327.56 ± 247.26	0.966	2.92 × 10^−5^
hsa-miR-525-5p ^a^	9.07 ± 15.31	0.952	1.63 × 10^−9^	4.21 × 10^−7^	79.89 ± 71.16	0.982	0.04471
hsa-miR-516a-5p ^a^	31.25 ± 53.81	0.959	4.55 × 10^−9^	7.83 × 10^−7^	150.92 ± 135.76	0.974	0.00141
hsa-miR-518e-5p ^a^	43.96 ± 62.52	0.962	8.92 × 10^−9^	1.22 × 10^−6^	99.07 ± 88.52	0.981	0.02118
hsa-miR-520a-3p ^a^	86.75 ± 132.37	0.960	9.41 × 10^−9^	1.22 × 10^−6^	220.91 ± 243.90	0.982	0.04009
hsa-miR-518e-3p ^a^	7.99 ± 14.22	0.956	2.02 × 10^−8^	2.09 × 10^−6^	29.60 ± 25.75	0.971	0.00189
hsa-miR-512-3p ^a^	287.04 ± 575.48	0.963	3.63 × 10^−8^	3.41 × 10^−6^	771.74 ± 982.11	0.982	0.02747
hsa-miR-1283 ^a^	57.10 ± 87.82	0.962	4.17 × 10^−8^	3.59 × 10^−6^	203.52 ± 173.84	0.977	0.00635
hsa-miR-517a-3p ^a^	20.19 ± 37.78	0.960	5.17 × 10^−8^	4.11 × 10^−6^	76.56 ± 73.88	0.980	0.01599
hsa-miR-526b-5p ^a^	17.12 ± 25.90	0.962	2.19 × 10^−7^	1.62 × 10^−5^	39.62 ± 34.83	0.977	0.01056
hsa-miR-517-5p ^a^	16.90 ± 29.75	0.962	9.56 × 10^−7^	6.17 × 10^−5^	70.67 ± 67.45	0.981	0.02191
hsa-miR-519c-3p ^a^	11.08 ± 20.08	0.965	5.11 × 10^−6^	0.00026	36.44 ± 32.83	0.966	0.00034
hsa-miR-519d-5p ^a^	6.12 ± 9.51	0.961	1.53 × 10^−5^	0.00063	10.75 ± 11.77	0.969	0.01205
hsa-miR-515-5p ^a^	10.50 ± 19.12	0.967	1.50 × 10^−5^	0.00063	26.70 ± 26.08	0.968	0.00076
hsa-miR-27b-3p	29,699.76 ± 20,347.78	1.008	4.98 × 10^−5^	0.00172	55,214.27 ± 41,098.16	1.011	0.00110
hsa-miR-885-5p	13.50 ± 17.92	1.036	4.98 × 10^−5^	0.00172	40.63 ± 78.98	1.034	0.00588
hsa-miR-520a-5p ^a^	3.06 ± 5.66	0.963	0.00032	0.00830	27.63 ± 27.09	0.965	0.00136
hsa-miR-375	1938.28 ± 3939.32	0.975	0.00040	0.00986	2900.08 ± 2878.58	0.982	0.03889
hsa-miR-520d-3p ^a^	9.80 ± 15.84	0.972	0.00091	0.01877	10.77 ± 13.37	0.966	0.00638
hsa-miR-592	1.64 ± 3.00	1.042	0.00162	0.03103	2.43 ± 4.89	1.058	0.00400
hsa-miR-21-5p	85,256.17 ± 61,108.55	1.005	0.00224	0.03811	82,095.57 ± 63,195.51	1.008	0.02738

Models adjusted for sequencing run and lane, gestational age, maternal age at the first trimester of pregnancy, and parity. * Fold changes represent the change in miRNA abundance for each increase of one unit of maternal BMI at 1st trimester of pregnancy. ^a^ miRNAs from C19MC. Abbreviations: 3D: Design, Develop, Discover birth cohort; FDR: false discovery rate; Gen3G: Genetics of Glucose regulation in Gestation and Growth birth cohort; SD: standard deviation.

## Data Availability

The datasets used and/or analyzed during the current study are available from the corresponding author upon reasonable request.

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
