# Peer review of "Maternal Body Mass Index Is Associated with Profile Variation in Circulating MicroRNAs at First Trimester of Pregnancy"

_biomedicines, 2022, doi:10.3390/biomedicines10071726_

Round 1

Reviewer 1 Report

There are only a handful of previous studies screening pregnancy plasma for biomarkers of disease risk e.g. gestational hypertension/PE, gestational diabetes. This particular study benefits from 2 cohorts, each with a considerable numbers of participants. Another strength is the -omics design. 

I appreciate that the data was adjusted for gestational age, maternal age and parity. However, seeing as GTT was a primary outcome in both cohorts, it  would be good to have a table of maternal characteristics stratifying or even adjusting the data for glucose tolerance AUC or insulin sensitivity index.

Minor comments

Although EPIC array data was acquired in one of the cohorts, no "epigenetic" data was presented, there I would remove the keyword "epigenetic".

Reviewer 2 Report

Comments,

This is an interesting study on the miRNAs expression in overweight/ obese pregnancy that could help in the comprehension of pathophysiology mechanisms in pregnancy.

 However, since fetal sex have a role in pregnancy outcome (Broere-Brown ZA,. Fetal sex and maternal pregnancy outcomes: a systematic review and meta-analysis. Biol Sex Differ. 2020 May 11;11(1):26. doi: 10.1186/s13293-020-00299-3.) and in miRNAs expression (Varì R et al. Significance of Sex Differences in ncRNAs Expression and Function in Pregnancy and Related Complications. Biomedicines. 2021 Oct 20;9(11):1509. doi: 10.3390/biomedicines9111509), and, taking into account the large number of subjects analyzed, it  would be interesting to compare, if possible,  miRNAs expression and fetal sex.
